# Development and Assessment of Regeneration Methods for Peptide-Based QCM Biosensors in VOCs Analysis Applications

**DOI:** 10.3390/bios12050309

**Published:** 2022-05-07

**Authors:** Tomasz Wasilewski, Bartosz Szulczyński, Dominik Dobrzyniewski, Weronika Jakubaszek, Jacek Gębicki, Wojciech Kamysz

**Affiliations:** 1Department of Inorganic Chemistry, Faculty of Pharmacy, Medical University of Gdańsk, Hallera 107, 80-416 Gdansk, Poland; weronika.jakubaszek@gumed.edu.pl (W.J.); kamysz@gumed.edu.pl (W.K.); 2Department of Process Engineering and Chemical Technology, Chemical Faculty, Gdańsk University of Technology, Gabriela Narutowicza 11/12, 80-233 Gdansk, Poland; bartosz.szulczynski@pg.edu.pl (B.S.); domdobrz@student.pg.edu.pl (D.D.); jacek.gebicki@pg.edu.pl (J.G.)

**Keywords:** sensors, biosensors, QCM, peptides, odorants, Piranha, plasma, cyclic voltammetry, biosensors fabrication

## Abstract

Cleaning a quartz crystal microbalance (QCM) plays a crucial role in the regeneration of its biosensors for reuse. Imprecise removal of a receptor layer from a transducer’s surface can lead to unsteady operation during measurements. This article compares three approaches to regeneration of the piezoelectric transducers using the electrochemical, oxygen plasma and Piranha solution methods. Optimization of the cleaning method allowed for evaluation of the influence of cleaning on the surface of regenerated biosensors. The effectiveness of cleaning the QCM transducers with a receptor layer in the form of a peptide with the KLLFDSLTDLKKKMSEC-NH_2_ sequence was described. Preliminary cleaning was tested for new electrodes to check the potential impact of the cleaning on deposition and the transducer’s operation parameters. The effectiveness of the cleaning was assessed via the measurement of a resonant frequency of the QCM transducers. Based on changes in the resonant frequency and the Sauerbrey equation, it was possible to evaluate the changes in mass adsorption on the transducer’s surface. Moreover, the morphology of the QCM transducer’s surface subjected to the selected cleaning techniques was presented with AFM imaging. The presented results confirm that each method is suitable for peptide-based biosensors cleaning. However, the most invasive seems to be the Piranha method, with the greatest decrease in performance after regeneration cycles (25% after three cycles). The presented techniques were evaluated for their efficiency with respect to a selected volatile compound, which in the future should allow reuse of the biosensors in particular applications, contributing to cost reduction and extension of the sensors’ lifetime.

## 1. Introduction

The development of biosensors technology is aimed at approaching their biological counterparts’ sensitivity as closely as possible [1,2]. Quartz Crystal Microbalances (QCMs), which belong to piezoelectric transducers operating on the piezoelectric effect principle [3], are built from a thin quartz plate of a suitable shape and thickness with gold electrodes deposited on both sides (Figure 1). They allow for detection of a change in the mass adsorbed on a gold electrode’s surface in real time, with an accuracy of up to ca. 0.1 ng. QCMs can operate in liquid and gas media and offer frequency measurement in the range from a few to tens of Hz. This is especially useful for characterization of the biosensing interfaces, which contain various types of receptor layers, or in the evaluation of molecules’ binding dynamics [4,5,6]. The resonant frequency of the QCM transducers depends on the thickness of the piezoelectric material; however, it must be emphasized that thinner plates are more fragile and susceptible to mechanical damages. The QCM surface can be covered with different coatings, such as metals, metal alloys, metal oxides, semiconductors or polymers [4,7]. In order to ensure suitable sensitivity and accuracy of measurements, it is important to develop not only a reproducible method for peptide immobilization on the QCM-type transducer’s surface [8], but also an effective method for cleaning gold electrodes [9]. A single sensor can be used multiple times, which requires efficient techniques for cleaning and regeneration of the quartz crystal microbalances. The electrodes with a suitably prepared surface can serve as a substrate for the preparation of self-assembled monolayers (SAMs). The monolayers constitute ordered aggregates of organic molecules formed via their adsorption from a solution in a regular array on a metal surface. Recovery of a clean electrode requires the effective removal of SAMs from its surface in a way that does not damage the gold working electrode. A high affinity of thiol molecules to the metal surface allows for the formation of well-defined surfaces with useful and tuneable chemical properties.

Alkanethiols are the most popular monolayers used in the modification of quartz crystal microbalances [10]. Thiol groups exhibit a high affinity to the gold-forming RS-Au bond, thus easily forming an ordered monolayer on a gold surface. By activating selected groups, it is possible to obtain a microbalance’s surface with the desired chemical activity, functionality and dimension. Synthetic peptides deposited in the form of a monolayer reveal the easy functionalization and polar character of the peptide bond providing additional stabilization to a structure [11]. Unfortunately, insufficient cleaning of the transducer prior to immobilization has a negative impact on the effectiveness of the monolayers’ deposition and can lead to a deterioration of the biosensors’ response parameters. Additionally, gold surfaces easily absorb impurities and contaminants during transport and storage, so the influence of cleaning on brand new microbalances was also tested. Good anchoring of the SAMs during deposition also requires prior proper cleaning of the secondary transducer’s surface. This paper compares three techniques for the cleaning and regeneration of quartz crystal microbalances: cleaning via the electrochemical, oxygen plasma and Piranha solution methods. Oxygen plasma is widely used in the removal of organic compounds and sulphur from the sensor’s surface. However, it can lead to surface oxidation and the formation of gold oxide (Au_2_O_3_), which is instable at room temperature. Piranha solution can also have a negative influence on the sensor’s surface leading to erosion, which can cause further problems with “reimmobilization” of the peptides on the transducer. Accordingly, it is important to develop cleaning techniques that will not cause serious damage to the sensor’s surface, thus allowing its multiple use.

Peptide-based biosensors are gaining huge attention in various applications, where selective and sensitive analysis of VOCs is necessary [12,13,14]. Moreover, numerous electrochemical biosensors employing peptides as biorecognition layers for the detection of analytes, such as metal ions, proteins, nucleic acids and enzymes, have been validated [15,16,17]. Recently, numerous (bio)sensors and their arrays, which grant sophisticated monitoring and detection of VOCs, have been effectively employed in various applications [12]; nevertheless, the global trend of sustainable development has forced the design of these type of instruments to include greener methods [18]. Based on this tendency, the techniques that are considered green (involving small amounts of less or non-hazardous reagents), electrochemical and plasma cleaning, were compared to the most popular one, the Piranha cleaning technique [19]. The aim of the paper is a comparison of commonly employed techniques for the cleaning and regeneration of QCM biosensors immobilized with a peptide solution. The advantages and disadvantages of the particular methods were determined based on the cleaning efficiency of particular biosensors and their post-cleaning operation parameters as the gas biosensors. Figure 1 shows a schematic representation of a peptide layer on the transducer’s surface.

## 2. Materials and Methods

Piezoelectric transducers with a quartz plate (13.7 mm in diameter) and polished gold surface (5.1 mm in diameter) were used for all experiments. The AT-Cut 10 MHz QCMs with Au electrodes were acquired from OpenQCM (Novaetech s.r.l., Napoli, Italy). The frequencies before and after sensing were measured using OpenQCM Software. The resonant frequency of the bare QCM transducers was measured with QpenQCM Wi 2 (Novaetech s.r.l., Napoli, Italy) and a previously developed system [20]. All reagents and volatiles were purchased from Sigma-Aldrich (Sigma Aldrich Co., St. Louis, MO, USA).

### 2.1. Peptide Synthesis and Deposition on QCM Transducers

The peptide with the KLLFDSLTDLKKKMSEC-NH_2_ sequence (OBPP4) was synthetized according to the established method [21], employing the solid-phase Fmoc/tBu strategy. Briefly, the synthesis was carried out automatically on an automated microwave peptide synthesizer (Liberty Blue™, CEM Corporation, Mathews, NC, USA). The peptides were purified by reversed-phase high-performance liquid chromatography (RP-HPLC) with LP-chrom software. The crude and purified peptides were analysed by HPLC in a water/acetonitrile gradient. The purity (>95%) was confirmed by HPLC/UV-VIS (Varian, Mulgrave, VIC, Australia), and the identity was confirmed by LC/MS (Waters Acquity SQD, Milford, MA, USA). The lyophilized peptides were kept in the dark at 5 °C before deposition. Stability of the biosensors’ layers were evaluated as previously described [20].

The purified peptide was deposited on the quartz crystal microbalances using a drop-casting technique. It is a reproducible, fast and easily-accessible technique employing a relatively low volume of coating solutions and leading to thin uniform layer [8]. The thickness of the obtained coating depends on the volume, degree of dispersion, concentration, properties of the solvent and angle of contact between the substrate and solvent. Then, the sensors were placed in a desiccator for 24 h. The peptide was immobilized due to the phenomenon of SAM formation on the gold surface. The peptide solution was prepared using deionized water as a solvent. The peptide with a concentration of 10 mg·mL^−1^ and a volume of 20 µL was deposited on the gold electrodes’ surfaces on the QCM transducers using the electronic pipette Eppendorf Xplorer (Eppendorf, Hamburg, Germany). Each deposition process was carried out at room temperature following the method, which had been optimized before [8,22]. All cleaning processes were preceded by cleavages with deionized water and methanol and dried in the desiccator for around 12 h.

When biomolecules such as peptides adsorb on the crystal, and thus increase the crystal thickness, the device will receive a frequency change response [23]. According to the literature data, the degree of deposition can be evaluated based on changes to the resonant frequency before and after peptide immobilization, and it is expressed in µg/mm^2^. Following the Sauerbrey Equation (1), it is possible to determine an exact change in the peptide mass bound with the sensor [24].
(1)$$\Delta F=\frac{-2F_{o}^{2}\Delta M}{\left[A{\left(\mu_{q}\rho_{q}\right)}^{\frac{1}{2}}\right]}$$
where, $\rho_{q}$ and ${µ}_{q}$ are the density (2.648 g·cm^−3^) and shear modulus of quartz (2.947 × 10^11^ g·cm^−1^·s^2^), respectively; $f_{0}$ is the crystal fundamental frequency of the piezoelectric quartz crystal; A is the crystal piezoelectrically active geometrical area, which is defined by the area of the metallic film deposited on the crystal; and Δm and Δf are the mass and frequency changes [23,24].

### 2.2. Piranha Cleaning

An important problem connected with application of the QCM electrodes is a dependence between the response of the system and the condition of the electrodes’ surfaces. Utilization of the QCM electrodes calls for precise cleaning in order to arrive at reproducible results. The method for cleaning and regeneration depends on the material of which the electrodes are made. One of the methods is the application of Piranha solution. It is prepared by mixing concentrated sulphuric acid and a 30% solution of hydrogen peroxide in the ratio 7:3. The frequency of QCM was measured, and then each sensor was immersed in 10 mL of Piranha solution for 10 min. Afterwards, the QCM was rinsed with demineralized water and ethanol, dried with nitrogen, and then the frequency was measured again. The literature reports additional heating of the solution (40–60 °C); however, Piranha solution intrinsically elevates its temperature to that level because the mixing of both its components triggers an exothermic reaction. Accordingly, heating the solution was excluded from the experiment schedule since it is not necessary for cleaning the QCM gold surfaces.

### 2.3. Oxygen Plasma Cleaning

Oxygen plasma can be employed for fast removal of contaminants from non-metallic surfaces, such as glass and plastic, as well as from metallic surfaces, e.g., gold, as in the discussed experiment. It does not react with silicon dioxide, so the quartz crystal’s properties do not change during cleaning of the microbalance. Organic contaminants are removed from a surface faster than inorganic ones due to a higher tendency for oxidation of the organic compounds. This method is also effective in the removal of sulphur from a surface [25,26]. Oxygen plasma cleaning of the quartz crystal microbalances is an easy, fast and environmentally friendly method. During the oxygen plasma treatment, the surface energy of the deposited polymers increases, which changes the hydrophobicity of the surface. Different oxygen functional groups, such as C–O, C=O, O–C=O and C–O–O, are formed on the surface of the quartz crystal microbalance. Etching of the polymer occurs as a result of the reaction between the atomic oxygen and carbon atoms on the surface, which produces a volatile product, such as CO [27]. Oxygen plasma strongly interacts with the magnetic and electric fields, and it is a good conductor. The principle of operation consists in the electric current flow through the plasma, which generates highly energetic electrons that collide with the gas molecules present in a reactor and break chemical bonds. Additionally, oxygen plasma causes oxidation of a cleaned surface, forming gold oxide (Au_2_O_3_). It does not damage or minimize the defects on a cleaned surface. Oxygen plasma contains both O^+^ and O^2+^ ions, so its molecules possess a lower momentum that is transferred to the surface, as compared to argon, thus accounting for lower mechanical erosion [26,28]. The QCM frequency was measured before and after plasma cleaning. Oxygen plasma cleaning was carried out in the plasma generator Atto Diener (Diener Electronic GmbH, Germany) for 85 s, at a pressure of 0.5 mBa and power equal to 22 W; the chamber volume was 2000 cm^3^. After cleaning, the QCM was placed in a beaker with deionized water for 3 min, and then transferred to methanol, followed by drying in a desiccator (12 h).

### 2.4. Electrochemical Cleaning

In application, electrochemical cleaning consists of a few oxidation/reduction cycles during a voltamperometric method. The surface morphology of the electrode influences the process of charge transfer at the interface [29]. Cleaning and regeneration of the microbalances was also performed with the technique based on a change in the potential [30,31]. The QCM electrodes were cleaned using a potentiostat/galvanostat (PGSTAT204, Metrohm, Utrecht, The Netherlands) in a three-electrode system in which Au was a working electrode, Pt was a counter electrode and Ag/AgCl served as a reference electrode. Three electrochemical treatments were used (Table 1).

In order to compare the presented methods and check the quality of electrode cleaning, each QCM was subjected to three CV cycles in probe solution (before and after cleaning). A total of 10 mM [Fe(CN)_6_]^3−/4−^ redox couple and 200 mM KCl electrolyte in water were used as the probe solution. The potential was swept from −100 to 700 mV (vs. reference electrode). As a parameter for the chemical cleanliness of an electrode’s surface, the potential difference between the anodic and cathodic peaks was subjected.

Each cleaning method was applied to three transducers for statistical purposes. Similar to previous methods, after a single cycle, cleavage with water and methanol was completed and then drying in a desiccator. The cleaning cycle for the electrochemical cleaning method was accomplished; Method no. 1 and no. 3 were accomplished in under 7 min and method no. 2 was accomplished in 4 min. In addition, the time required for sensor handling and assembling/disassembling into an electrochemical cell needs to be taken into account (around 2–3 min). Electrochemical cleaning with a used apparatus was restricted to cleaning a single sensor in one cycle.

### 2.5. Surface Characterization

The surface topography of the QCM transducers was characterized using an atomic force microscope (AFM Ntegra Prima, NT-MDT, Moscow, Russia) employing the NSG 01 probes. AFM images of the surface were collected in order to obtain the characteristics of cleaned surfaces on the sensors. This measurement does not require special sample preparation or special ambient conditions. The measurements were conducted in a tapping mode. The geometrical dimensions of the probe were 125 × 30 × 2; its resonant frequency was equal to 150 kHz, and its spring constant amounted to 5.1 N/m.

### 2.6. Measurement Setup

The biosensor with the OBPP4 (peptide mimicking HarmOBP7 “binding pocket” region) receptor element was characterized in detail in a previous paper that revealed the lowest limit of detection (LOD) for nonanal: 14 ppm. That is why a reference for this gas substance was used to evaluate the performance of the biosensors regenerated after deposition cycles. Nonanal, at a concentration of 65 ppm, was prepared in Tedlar^®^ bags using a gas mixture generator [20]. Previously developed systems were used for the gas mixture’s generation and biosensors’ measurements [8]. The correctness of their preparation was verified with a gas chromatograph (430-GC, Bruker^®^, Bremen, Germany) according to the method elaborated earlier [20]. The PTFE chamber (65 cm^3^) with a peptide-based biosensor inside was saturated using a low-pressure pump system. Pure air (2 ± 1% relative humidity) was utilized as a carrier gas, employed for desorption of volatile analytes. The response of a detector, which indicated adsorption of the gas molecules in a receptor part, was a change in frequency with an accuracy of ±1 Hz. The processing and archiving of data was accomplished with dedicated QCMmeter software (created by the Gdansk University of Technology). The total measurement time was unified to 300 following the protocol presented before [22]. The degree of adsorption of the given ligands determined the time necessary for sorption/desorption. A baseline for the sensors (F_I_) was established by flushing dry air, and exposing the sensors to a specific concentration of nonanal. After the introduction of the gaseous phase, the sensor frequency was reduced until the steady state was reached due to maximum adsorption of gas molecules (F_II_). Finally, a return to the initial sensor baseline was achieved by replacing the nonanal gas with pure nitrogen gas. Differences between the recorded F_i_ and F_II_ were calculated accordingly. Three measurements were used to obtain the average values of the biosensors’ responses. To confirm the negative responses, the response of a bare QCM electrode was monitored prior all measurements.

## 3. Results

The possibility of biosensors’ regeneration via suitable cleaning techniques was analysed for a new, bare transducer, as well as for regenerated ones after deposition of the receptor layers (synthetic peptide). The preliminary cleaning was tested on new, intact electrodes to check the potential impact of cleaning on later parameters of deposition and transducer operation. The degree of deposition was evaluated by observation of the resonant frequency changes measured before and after peptide immobilization.

### 3.1. Bare Gold Electrodes Cleaning

Cleaning of the bare electrodes consisted of measurement of the frequency of four brand-new sensors, followed by three cleaning cycles with the electrochemical (sensors no. QCM1, QCM2 and QCM3), plasma (sensors no. QCM4, QCM5 and QCM6) and Piranha solution (sensors no. QCM7, QCM8 and no. QCM9) methods. The resonant frequency of the sensors was measured again after cleaning and drying. Obtained differences are presented in Figure below (Figure 2).

In the cases of all transducers, there is an increase in the resonant frequency after three cleaning cycles. Following the Sauerbrey Equation (1), this increase in the resonant frequency of the quartz crystal microbalance corresponds to a mass decrement in the sensor, which is related to erosion of the electrodes’ surfaces due to applied cleaning. The biggest increase was observed for the sensors regenerated with the Piranha solution, so it can be assumed that this type of cleaning interferes with an electrode’s surface much more strongly than in case of the other two cleaning techniques. A change in the sensors’ frequency after deposition is different for each electrode because the Piranha solution imposes a different impact on various places on a gold surface. Regarding the oxygen plasma method, the biggest mass decrement can be noticed during the first cleaning cycle. Electrochemical cleaning reveals the best reproducibility and leads to the smallest losses in the structure of transducer’s gold layer.

### 3.2. Cleaning after Deposition

Prior to and between the cleaning cycles, all sensors were immobilized using the drop-casting technique on one part of the electrode. Cleaning of the electrodes consisted of measurement of the resonant frequency of bare sensors, deposition of peptide and an application of three cleaning cycles with the selected techniques. After cleaning and drying, the resonant frequency of the sensors was measured again, and a sensor’s mass related to a sensor’s surface was calculated (according to the Equation (1)). The results are presented in Table 2 and Figure 3.

For all regeneration methods, a greater frequency change was observed during cleaning than during peptide deposition. This supports the conclusion that each of the methods not only removes the deposited layer of the peptide, but also removes some of the other impurities that are on the electrode surface. In the case of the electrochemical method, this change is the smallest, which allows us to conclude that it is the mildest method for cleaning the electrode surface. Sensors no. D1, D2 and D3 were purified by various electrochemical methods; therefore, it is possible to compare the average frequency change for each of the methods: HCl solution—53.7 Hz, H_2_SO_4_ solution—8.0 Hz and KOH solution 43.3 Hz. As can be seen, electrochemical treatment with a sulphuric acid solution allows for the most selective removal of the deposited peptide layer without disturbing the electrode structure (as well as its impurities) prior to deposition.

The cleanliness of the electrode surface was also assessed with the use of cyclic voltammetry (CV) in a three-electrode system. The example of CV in a probe solution before and after the potential cycling cleaning (in sulphuric acid) is presented in Figure 4. It also shows the values of the anodic and cathodic peaks, as well as the potential difference for the sensor after immobilization and cleaning.

Table 3 shows the change in potential as a percent difference (%Δ) from their original, uncleaned sample values.

Theoretically, for single-electron transfer reactions, such as in the ferri/ferro-cyanide couple on a perfect gold surface, the potential difference (ΔE_p_) should be equal 57 mV [32]. 

It is challenging to obtain a value close to the theoretical one. It is mainly influenced by: the quality of the electrode, the speed of voltage changes and the distance between the electrodes in the system. However, in the case of constant measurement conditions, the changes observed in this parameter concern only the quality of the electrode surface. We assume that any increase in this value is caused by surface imperfections or contamination (in this case shown by a layer of deposited peptide). The highest value of the potential difference change is shown in the potassium hydroxide potential sweep, which proves that this method removes the most contamination from the electrode surface. However, the cleanest surface is obtained after cleaning with the hydrochloric acid potential cycling method (the lowest ΔE_p,clean_ values, closest to the 57 mV value).

### 3.3. AFM Analysis

Figure 5A illustrates the image of a peptide layer deposited on the QCM transducer. The characteristics of deposited layers depending on applied deposition technique and deposition parameters were presented in details in the paper [8]. Analysis of obtained AFM images shows that the technique utilizing Piranha solution leads to an increase in surface roughness of the sensor’s surface due to erosion. In Figure 5D there are discolorations indicating microdefects of the surface. It means that polished structure of the gold electrode was damaged. In Figure 5A,B there are no such defects or they appear sporadically (Figure 5C). That is why one can conclude that oxygen plasma and electrochemical cleaning are less invasive and impose lower degradation of the gold surface. It can be connected with the fact that oxygen plasma cleaning leads to surface oxidation of cleaned electrode and formation of protective gold oxide layer (Au_2_O_3_).

### 3.4. Biosensors Responses to Gaseous Compounds

As proved in earlier studies [24], the biosensor with the OBPP4 active element reveals a high affinity to long-chain aldehydes, octanal, nonanal and undecanal. The biosensors immobilized with the selected peptide were tested with respect to a reference gas—nonanal. For further testing of the biosensors against the nonanal regenerated by electrochemical methods, the technique in which potassium hydroxide was used was chosen, which was associated with the highest repeatability and the least surface degradation during purification cycles with this treatment technique. The OBPP4 peptide was deposited on bare sensors, and then three measurements upon exposure to the reference gas were performed. Afterwards, the sensor’s surface was regenerated using one of the three methods, and the biosensor was tested again with respect to the selected reference gas. The biggest decrease in sensitivity was observed for the biosensors regenerated with the Piranha solution. The difference in the biosensor response to nonanal before the first deposition cycle and after the third deposition cycle was 17 Hz (±1 Hz), which corresponds to a 25% drop in performance. A smaller decrease in performance was recorded for the biosensors cleaned via the electrochemical and oxygen plasma methods at 7% and 16%, respectively.

## 4. Discussion

When cleaning bare QCM electrodes with Piranha solution, the highest frequency change can be observed (Figure 2C). This results in a decrease in the possible active area for a transducer to effectively form SAMs. A decrease in the active layer potentially results in a loss of biosensors’ sensitivity (Figure 6C). Moreover, a decrease in the active area can be observed when the biosensors with SAMs are regenerated (Figure 3). Similarly, the biggest drop in the frequency change can be observed for the biosensors regenerating with Piranha solution (Figure 3C). Furthermore, regeneration with Piranha solution can lead to a serious change in the physical properties of the passivated gold surface, which may require salinization in order to achieve the adequate surface properties. This is due to the fact that treatment with the Piranha solution causes a change in the surface wettability [33]. Silanization enables better retention of the droplets used during immobilization, making this process more accurate and repeatable. Electrochemical cleaning (Figure 2A) results in the lowest frequency change (3–16 Hz) and confirms that this technique is the least invasive for gold electrodes. For sensors no. B2, B3 and C3 (Figure 3), cleaned with plasma and Piranha, respectively, it can be observed that the first cleaning cycle results in an incomplete removal of the receptor layer from the biosensors. This could be a consequence of the challenge of achieving reproducibility for the Piranha solution technique. Further, the effect of the operation of instable plasma can occur during regeneration cycles. Considering the frequency change values shown in Figure 3, the electrochemical cleaning technique appears to be the most reproducible. For all regeneration cycles with the electrochemical method (Figure 3A), the frequency change remains at a comparable level, which indicates a good removal of SAMs, while keeping the gold surface almost intact and ready for further depositions. The third cycle of cleaning with the potential hydrochloric acid and sulphuric acid cycling treatments results in a slight increase in the frequency for sensors no. QCM1 and QCM2 (Figure 3A), as a consequence of the higher frequency change (12–16 Hz). In addition, this negligibly affects the biosensor performance in the case of sensitivity loss (Figure 6A). A potassium hydroxide potential sweep provides the most reproducible cleaning of bare electrodes (Figure 2A, sensor no. QCM3).

The images of the electrodes’ surfaces, collected using the atomic force microscope, show significant microdefects of the surface of the transducers cleaned with the Piranha solution. The smaller number of defects in the case of the oxygen plasma method could be the result of the formation of protective gold oxide (Au_2_O_3_). The electrochemical technique seems to be the least invasive approach to cleaning the QCM electrodes, allowing for extension of the sensor’s lifetime. This was confirmed by measuring the frequency change in the sensor before and after cleaning. These observations are unequivocally supported by the biosensors’ performance after the cleaning cycles where the biggest decrease in performance is observed for the biosensors that were regenerated with the Piranha method (25%); a lower decrease can be noted for the oxygen plasma technique (16%). The oxygen plasma treatment can also lead to oxidation of the cleaned surface and cause formation of oxides on an electrode’s surface. This can make it troublesome to covalently attach the peptide to the electrode surface. Such a phenomenon was observed in the presented research—a decrease in the mass of the attached peptide was recorded in three consecutive deposition cycles. The influence of a degree of electrode oxidation due to this particular regeneration technique will be the subject of further investigations. Application of the Piranha solution and oxygen plasma results in very efficient cleaning of the surface; however, it changes the physical properties of the sensor and causes erosion of gold, thus decreasing the effectiveness of “reimmobilization” and the sensor’s performance.

## 5. Conclusions

The investigations conducted within this study were aimed at evaluation of the effectiveness of selected cleaning and regeneration techniques for the peptide-based biosensors, employing a quartz crystal microbalance with a gold electrode surface as a transducer element. A well-cleaned electrode surface on the QCM transducer is indispensable for obtaining densely-packed, and thus functional, SAMs. That is why it is of the upmost importance to precisely clean the biosensor’s surface where SAMs are deposited. Properties of the monolayers deposited on the piezoelectric transducers are important from the standpoint of the biosensors’ design and metrological parameters. The presence of defects on the surface of electrodes ready for immobilization significantly influences the biosensors response to selected analytes. A decrease in the biosensors’ sensitivity can be noticed after successive cleaning cycles, which leads to less intensive signals as far as changes in the QCM biosensors’ frequency are concerned. A summary of the pros and cons of the three tested methods is presented in Table 4.

All applied cleaning techniques offer the removal of covalently bonded peptides from the gold electrode surface of QCMs. In the case of the technique with the Piranha solution, there is the biggest mass decrement, which indicates that the technique influences erosion of the electrode’s surface. Deterioration of the performance of the sensors treated with the Piranha method may be also connected with the fact that both cleaning methods change the wettability of the sensor’s surface, which can yield further problems with “reimmobilization” of the receptor layer. Each technique reduces the active surface area of the electrode, contributing to a decrease in immobilisation efficiency and, consequently, a decrease in sensitivity after subsequent purification cycles. This problem is most significant in the case of purification with the Piranha solution; plasma-based purification damages the electrode surface to a much lower extent. In contrast, all variants of electrochemical cleaning cause little damage; this is potentially the least invasive technique for cleaning pure gold QCM electrodes and regenerating biosensors with an applied SAM layer. The electrochemical method turned out to be the least invasive. In the case of purification in sulphuric acid solution, it allows for the most selective removal of the deposited peptide layer. It must be also emphasized that the electrochemical and plasma cleaning techniques are much more environmentally friendly than the Piranha solution method.

## Figures and Tables

**Figure 1 biosensors-12-00309-f001:**
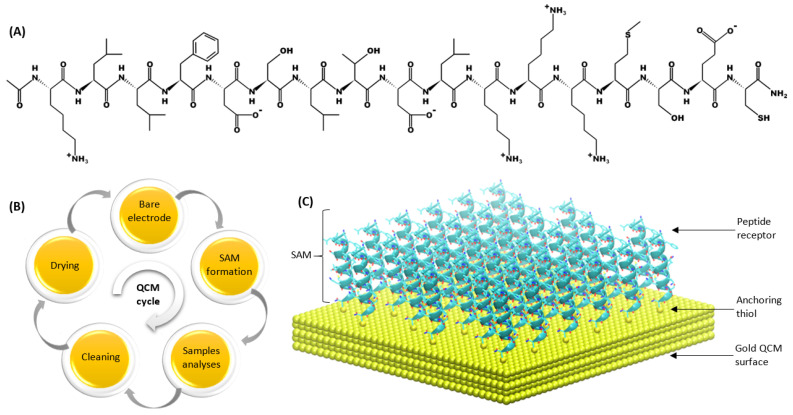
(**A**) Structure of peptide (KLLFDSLTDLKKKMSEC-NH_2_) used for SAM formation; (**B**) diagram of peptide-based QCM biosensor life-cycles; (**C**) schematic organization of a peptide SAM on one side of the gold QCM electrode.

**Figure 2 biosensors-12-00309-f002:**
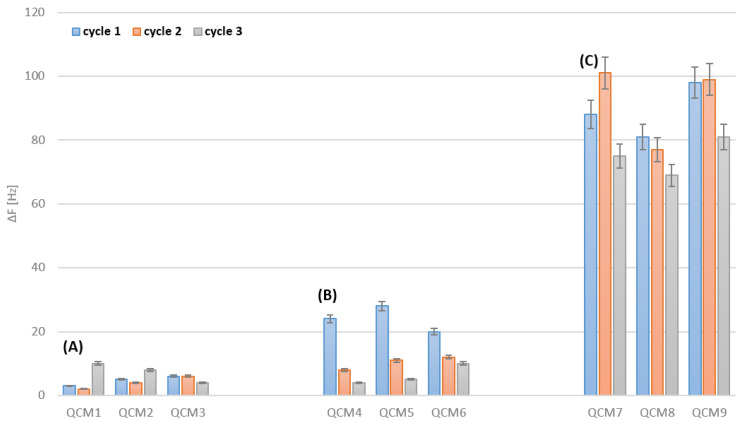
Changes in resonant frequencies [Hz] of bare gold QCM electrodes after regeneration: (**A**) electrochemical (QCM1—hydrochloric acid potential cycling, QCM2—sulphuric acid potential cycling, QCM3—potassium hydroxide potential sweep), (**B**) plasma and (**C**) Piranha cleaning methods. Error bar indicates the calculated standard deviation (n − 3).

**Figure 3 biosensors-12-00309-f003:**
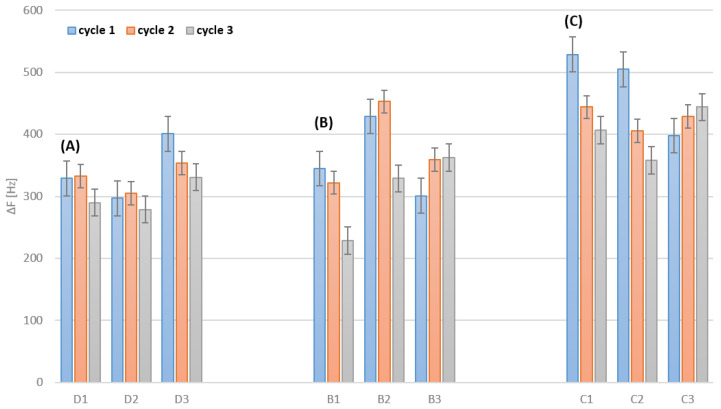
Changes in resonant frequencies [Hz] of QCM electrodes with peptide-based films after different cleaning techniques: (**A**) electrochemical (D1—hydrochloric acid potential cycling, D2—sulphuric acid potential cycling, D3—potassium hydroxide potential sweep), (**B**) plasma and (**C**) Piranha. Error bar indicates the calculated standard deviation (n − 3).

**Figure 4 biosensors-12-00309-f004:**
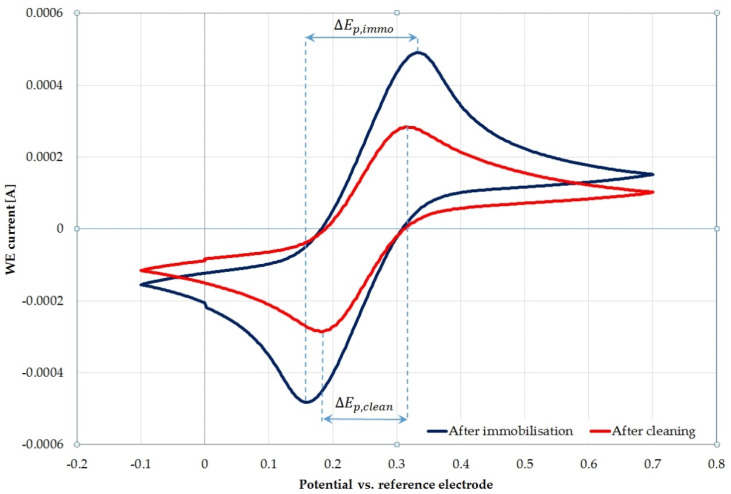
Comparison of CV scans before and after potential cycling cleaning (in sulphuric acid).

**Figure 5 biosensors-12-00309-f005:**
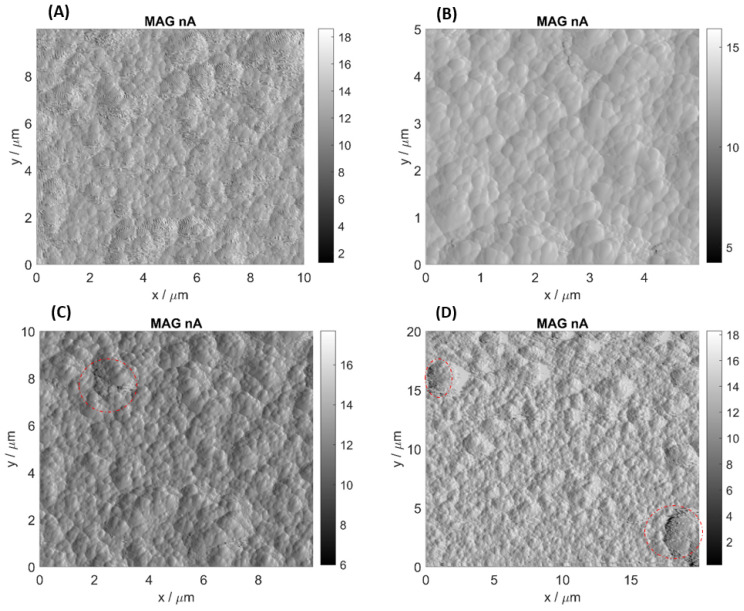
AFM images of QCM resonators: (**A**) with deposited OBPP4 and after (**B**) electrochemical cleaning, (**C**) plasma cleaning and (**D**) Piranha cleaning (surface defects are highlighted in red).

**Figure 6 biosensors-12-00309-f006:**
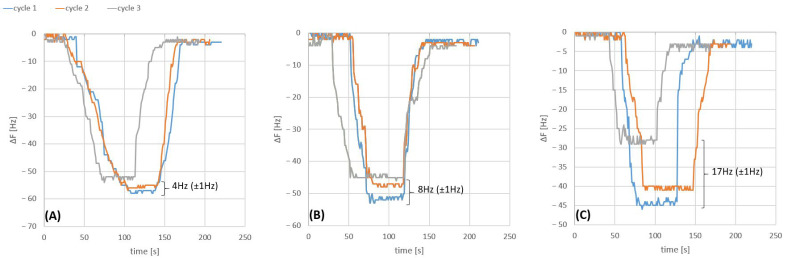
Resonant frequency responses of OBPP-4 based biosensors for nonanal in 65 ppm concentration. Measurements were performed after 3 cycles of cleaning with different methods: (**A**) electrochemical (potassium hydroxide potential sweep, sensor no. D3), (**B**) plasma (sensor no. B2), (**C**) Piranha (sensor no. C2).

**Table 1 biosensors-12-00309-t001:** Parameters of electrochemical treatment methods.

No.	Method	Cycle Range (vs. Reference Electrode)	Number of CV Cycles	Scan Rate	Solution	Total Time
1.	Hydrochloric acid potential cycling	−500 to 1500 mV	10	100 mV/s	50 mM HCl	400 s
2.	Sulphuric acid potential cycling	−400 to 1400 mV	12	100 mV/s	50 mM Sulphuric acid	240 s
3.	Potassium hydroxide potential sweep	−100 to −1200 mV	10	50 mV/s	50 mM KOH	440 s

**Table 2 biosensors-12-00309-t002:** Response of QCM transducers after OBPP4 peptide immobilization and after cleaning with selected techniques.

Sensor No.	Frequency Change after Immobilization [Hz], Mass Change [µg/cm^2^]	Frequency Change after Cleaning [Hz], Mass Change [µg/cm^2^]
	Cycle 1, 2, 3	Cycle 1, 2, 3
**Electrochemical**		
D1	−276 (+1214), −255 (+1122), −260 (+1144)	329 (−1448), 333 (−1465), 290 (−1276)
D2	−291 (+1280), −285 (+1254), −281 (+1236)	297 (−1307), 305 (−1342), 279 (−1228)
D3	−333 (+1465),−311 (+1368), −309 (+1360)	401 (−1764), 354 (−1558), 331 (−1456)
**Plasma**		
B1	−325 (+1430), −301 (+1324), −291 (+1280)	345 (−1518), 322 (−1417), 229 (−1008)
B2	−377 (+1659), −351 (+1544), −300 (+1320)	429 (−1888), 453 (−1993), 329 (−1448)
B3	−298 (+1311), −251 (+1104), −239 (+1052)	301 (−1456), 359 (−1580), 363 (−1597)
**Piranha**		
C1	−401 (+1764), −389 (+1712), −314 (+1382)	529 (−2328), 444 (−1954), 407 (−1791)
C2	−388 (+1707), −401 (+1764), −312 (+1373)	505 (−2222), 403 (−1773), 358 (−1575)
C3	−371 (+1632), −388 (+1707), −301 (+1324)	398 (−1751), 429 (−1888), 444 (−1954)

**Table 3 biosensors-12-00309-t003:** Changes in potential as a percent difference (%Δ) from their original, uncleaned sample values.

Sensor No.	Cycle	ΔE_p,immo_	ΔE_p,clean_	%Δ	%Δ_mean_
D1 (HCl cleaning)	1	0.14155	0.10786	−23.8	−22.0 ± 3.5
	2	0.13223	0.09891	−25.2	
	3	0.13755	0.11403	−17.1	
D2 (H_2_SO_4_ cleaning)	1	0.16389	0.13183	−19.6	−16.9 ± 2.7
	2	0.14405	0.11841	−17.8	
	3	0.13577	0.11785	−13.2	
D3 (KOH cleaning)	1	0.16630	0.10228	−38.5	−31.3 ± 5.2
	2	0.16079	0.11432	−28.9	
	3	0.16032	0.11800	−26.4	

**Table 4 biosensors-12-00309-t004:** A summary of the advantages and disadvantages of methods used in this study to clean and regenerate piezoelectric biosensors with the peptide (OBPP4) as receptor element.

Cleaning Technique	Advantages	Disadvantages
Piranha solution	Easy to handle, possibility to clean multiple sensors in single cycle, expensive instruments or reagents are not required	Very toxic, requires safety procedures, leads to surface erosion, significantly reduces sensors’ lifetime and sensitivity after multiple cleaning cycles, changes sensors’ surface wettability
Plasma cleaning	Possibility to clean multiple sensors in single cycle, use of expensive or toxic chemicals is eliminated, high control and repeatability	Slightly reduces sensors’ lifetime and sensitivity after multiple cleaning cycles, minor problems with correct plasma generation, expensive instrument, changes sensors’ surface wettability
Electrochemical cleaning	Insignificantly reduces sensors’ lifetime and sensitivity after multiple cleaning cycles, non-invasive for sensors’ surfaces, lower consumption of toxic reagents, safe and environmentally friendly, high control and repeatability	Single sensor can be cleaned in one cycle, time-consuming, complicated instrumentation that requires trained personnel, expensive instrument

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
