# Peer review of "Development and Assessment of Regeneration Methods for Peptide-Based QCM Biosensors in VOCs Analysis Applications"

_biosensors, 2022, doi:10.3390/bios12050309_

Round 1
Reviewer 1 Report
This paper studies the cleaning and regeneration methods of QCM sensors and compares the advantages and disadvantages of the three methods from the theoretical and experimental levels. In general, the author has done a lot of analysis work, and it is of certain significance. But I have some major concerns about the paper.
- Line 238-239: some sensors were wrongly numbered. The format can be unified to make it easier for readers to understand.
- Line 276-277: The number of the table should be Table 2. The data in the two tables were not further analyzed. The data source of 59mV is not explained. Why the potential difference (ΔEp) should be equal 59 mV in the ferri/ferro-cyanide couple on a perfect gold surface? Perhaps you can supplement this with references to relevant literature.
- According to Figure 2, you can only get the conclusion that "In the case of all transducers, there is an increase in the resonant frequency after 3 cleaning cycles. And the treatment method of piranha solution leads to the largest increase in frequency". However, the frequency variation trend and rule of the 3 cleaning cycles of the same electrode have not been summarized and explained. Is there any reasonable explanation? There is also no detailed chart analysis in Figure 3 either.
- For the comparison of the three regeneration methods, do these regeneration experiments carried out at the same time? What is the best regeneration time for the three methods? Which one is the fastest?
Author Response
Response to the Reviewer in the attached file.

Reviewer 2 Report
Wasilewski et. al reported work "Development and assessment of regeneration methods for peptide-based QCM biosensor in VOCs analysis applications" is interesting and well written. However this manuscript need minor revision before it get published in the Biosensors
1) Authors neglected some of the recent references specifically peptide based electrochemical sensors for VOCs in the introduction part.
2) In current work author demonstrated long chain aldehydes, only, so why only aldehydes? how about other materials such amines (biogenic amines are more predominantly found in biological systems.
3) Conclusion need to be separated from the discussion to avoid ambiguity for readers.
4) Some of the figure legends not clear it need to be coloured in black and bold font.
5) Some of the typographical mistakes need to be corrected.
Author Response

(The authors gave the same response as above.)

Round 2
Reviewer 1 Report
the chapter titles of the second and third chapters are very confusing and there are some careless numbering errors that need to be corrected. Here are the titles needed to be changed.
- Materials and Methods line 98
2.1. Peptide synthesis and deposition on QCM transducers line 106
2.3. Piranha cleaning line142
2.4. Oxygen plasma cleaning line156
2.5. Electrochemical cleaning line 182
2.6. Surface characterization line 206
2.6. Measurement setup line 214
3.2. Bare gold electrodes cleaning line 245
3.2. Cleaning after deposition line 268
3.2. AFM analysis line 315
3.2. Biosensors responses to gaseous compounds line 330

Author Response
Dear Editor,
Dear Reviewer,
We would like to express gratitude for fast and professional handling of our manuscript submitted to Biosensors. Minor changes suggested by the Reviewer were introduced in the manuscript.
Reviewer #1:
After the last revision, the article has been more complete, and some data and charts have been supplemented. However, the chapter titles of the second and third chapters are very confusing and there are some careless numbering errors that need to be corrected.
- Materials and Methods line 98
2.1. Peptide synthesis and deposition on QCM transducers line 106
2.3. Piranha cleaning line142
2.4. Oxygen plasma cleaning line156
2.5. Electrochemical cleaning line 182
2.6. Surface characterization line 206
2.6. Measurement setup line 214
3.2. Bare gold electrodes cleaning line 245
3.2. Cleaning after deposition line 268
3.2. AFM analysis line 315
3.2. Biosensors responses to gaseous compounds line 330
Response: According to the comment, numbering errors were corrected in the final version. Also, Figure 4 was corrected.
We hope our revision improve the paper to a satisfactory level. The Authors.
